# A Remote Sensing Approach for Assessing Daily Cumulative Evapotranspiration Integral in Wheat Genotype Screening for Drought Adaptation

**DOI:** 10.3390/plants12223871

**Published:** 2023-11-16

**Authors:** David Gómez-Candón, Joaquim Bellvert, Ana Pelechá, Marta S. Lopes

**Affiliations:** 1Efficient Use of Water in Agriculture Program, Institute of Agrifood Research and Technology (IRTA), Fruitcentre, Parc AgroBiotech, 25003 Lleida, Spain; joaquim.bellvert@irta.cat (J.B.); ana.pelecha@irta.cat (A.P.); 2Field Crops Program, Institute for Food and Agricultural Research and Technology (IRTA), 251981 Lleida, Spain; marta.dasilva@irta.cat

**Keywords:** water productivity, unmanned aerial vehicle, evapotranspiration, TSEB model, leaf area index

## Abstract

This study considers critical aspects of water management and crop productivity in wheat cultivation, specifically examining the daily cumulative actual evapotranspiration (ETa). Traditionally, ETa surface energy balance models have provided estimates at discrete time points, lacking a holistic integrated approach. Field trials were conducted with 22 distinct wheat varieties, grown under both irrigated and rainfed conditions over a two-year span. Leaf area index prediction was enhanced through a robust multiple regression model, incorporating data acquired from an unmanned aerial vehicle using an RGB sensor, and resulting in a predictive model with an R^2^ value of 0.85. For estimation of the daily cumulative ETa integral, an integrated approach involving remote sensing and energy balance models was adopted. An examination of the relationships between crop yield and evapotranspiration (ETa), while considering factors like year, irrigation methods, and wheat cultivars, unveiled a pronounced positive asymptotic pattern. This suggests the presence of a threshold beyond which additional water application does not significantly enhance crop yield. However, a genetic analysis of the 22 wheat varieties showed no correlation between ETa and yield. This implies opportunities for selecting resource-efficient wheat varieties while minimizing water use. Significantly, substantial disparities in water productivity among the tested wheat varieties indicate the possibility of intentionally choosing lines that can optimize grain production while minimizing water usage within breeding programs. The results of this research lay the foundation for the development of resource-efficient agricultural practices and the cultivation of crop varieties finely attuned to water-scarce regions.

## 1. Introduction

As the global population continues to grow, ensuring food security in the face of climate change and resource limitations has become a critical challenge. Agriculture is potentially the sector most affected by climate change, with more frequent droughts, floodings, and extreme-heat events [1]. Breeding plans and field phenotyping play crucial roles in the development of resilient crop varieties capable of withstanding these challenges. These plans involve selecting and crossing plant varieties to develop improved lines with desired traits. In the context of drought and water-use efficiency, researchers have extensively studied genetic diversity and identified promising genes and genomic regions associated with these traits. For example, some studies have identified key genetic markers linked to drought tolerance and water-use efficiency in maize and rice [2,3]. Incorporating such genetic information through breeding plans enables the development of drought-tolerant and water-efficient crop varieties.

Field phenotyping refers to the assessment and quantification of plant traits related to drought tolerance and water-use efficiency under natural field conditions. Remote sensing, proximal sensing, and physiological measurements are important high-throughput phenotyping tools for the evaluation of crop performance in water-limited environments [4]. Such tools allow for breeders to accurately identify and select plants with drought tolerance and higher water-use efficiency. Researchers have highlighted the importance of incorporating heat and drought stress tolerance varieties in breeding programs [5]. By utilizing breeding plans and field phenotyping, it is possible to identify and select traits that enhance crop resilience to rising temperatures, altered precipitation patterns, and other climate change-induced stresses.

High-throughput field phenotyping (HTFP) techniques play a crucial role in breeding programs aimed at addressing climate change effects in agriculture [6,7]. Among these techniques, remote sensing has emerged as a powerful tool for the non-destructive and large-scale monitoring of crop performance and stress responses. Remote sensing utilizes sensors on board satellites or airborne or unmanned aerial vehicles (UAVs) to capture spectral information, enabling the assessment of vegetation biophysical variables and crop performance at various spatial and temporal scales. For instance, studies on beans using hyperspectral imagery have demonstrated its effectiveness in quantifying plant water status and predicting crop yield under drought conditions [8]. Other studies have been conducted to evaluate the spectral assessment of drought tolerance in wheat genotypes under arid conditions [9,10]. Similarly, satellite-based remote sensing was used to monitor water-use efficiency in olive trees, highlighting its potential for large-scale assessment and management of water resources [11]. HTFP through remote sensing offers several advantages over traditional phenotyping methods. First, remote sensing provides rapid and non-destructive measurements, enabling the continuous monitoring of crop performance throughout the growing season. This real-time information facilitates the early identification of stress-related responses, allowing for breeders to select the most promising genotypes for further development. Second, remote sensing provides spatially explicit data, capturing variability across fields and enabling precision agriculture practices. This spatial information assists in identifying specific areas within fields that are more susceptible to water stress or exhibit varying water-use efficiency, aiding targeted interventions and resource allocation.

HTFP thermography has been employed to assess water stress in various crops [12,13]. UAVs equipped with thermal cameras constitute a powerful tool for the capture of high-resolution images and the detection of crop water stress [14,15]. Some authors used UAV-sensed thermal imagery to assess water stress in wheat, highlighting the potential of multitemporal images for the evaluation of wheat drought resistance [16]. However, evaluating actual plant transpiration in multiple plots has been challenging, and surrogate traits have been predominantly used due to its complexity.

In this regard, the use of thermal-based models has been widely used to estimate actual evapotranspiration (or latent heat flux) [17,18,19]. For instance, Brenner et al. [20] conducted research on grasslands and crops, highlighting the potential of thermal-based ET estimation methods for the monitoring of water-use efficiency. These methods allow for the accurate and non-destructive assessment of crop water requirements, aiding in water management decisions and resource optimization. Surface energy balance models, specifically using very-high-resolution thermal imagery from UAVs, have proven valuable for the HTFP of actual crop evapotranspiration (ETa) in breeding programs [21].

The two-source energy balance (TSEB) model is an advanced energy balance approach that integrates biophysical processes and thermal remote sensing data to estimate crop ETa and its partition components separately [18,19]. In cases where high-resolution thermal imagery is available and soil surface and canopy temperatures can be obtained directly, the model can achieve greater accuracies by applying a contextual approach [17,22]. For instance, some studies have applied the TSEB model with very-high-resolution UAV imagery in vineyards in order to quantify vine water requirements and assess ET partitioning [17,23,24]. The TSEB modelling scheme was also applied in an almond rootstock collection in order to evaluate differences in transpiration and water productivity [25]. This modeling approach allows for a comprehensive understanding of the water balance dynamics within agricultural systems, aiding efficient water use and resilience to water scarcity. To the best of our knowledge, only Gómez-Candón et al. [21] have used this methodology to assess differences in ETa in a set of durum wheat varieties under different irrigation regimes.

Remote sensing only provides information for a specific date corresponding to image acquisition. However, from an agronomic point of view, and with the objective of evaluating the water productivity functions of different varieties, it is interesting to obtain continuous information on plant water use throughout the entire growing season. Therefore, a methodology capable of interpolating information among image acquisition dates could contribute to the analysis and comparison of differences in ETa and water status among varieties over the growing season.

This study aims to achieve the following two objectives: (1) to introduce a comprehensive TSEB-based methodology for the estimation of seasonal ETa evolution, and (2) to demonstrate the practicality of the proposed methodology to assess variations in water productivity (estimated as the ratio between yield and cumulative ETa) in a collection of wheat varieties under different water regimes.

## 2. Results

### 2.1. Remote Sensing-Based Estimations: Leaf Area Index and Actual Crop Evapotranspiration

A multiple regression model was used to estimate LAI (Leaf Area Index) using VARI (Visible Atmospherically Resistant Index), PH (crop height), and f_c_ (fractional cover) as independent variables. The obtained equation was:(1)$$LAI=5.06PH+5.45VARI-1.66f_{c}-1.07,$$

The parameters of the multiple regression model are shown in Table 1. The relationship between the observed and estimated LAI had an RMSE (root mean square error) of 0.46 and an R^2^ of 0.85 (Appendix A). Equation (1) was employed to calculate the LAI for each variety and image acquisition date. Mean values of the VARI exhibited a consistent increase over the course of the cropping season for both treatment groups (as illustrated in Appendix A). The validation of remote sensing estimates for PH and LAI yielded significant results, with an R^2^ value of 0.99 and an RMSE of 0.01 m for PH, as well as an R^2^ of 0.85 and an RMSE of 0.46 for LAI (as shown in Appendix A). Figure 1 shows cumulative LAI, potential evapotranspiration (ETp), and ETa across all irrigation treatments and varieties. In 2021, maximum LAI values ranged from 4.6 to 9.6 for 100% ETc and from 4.6 to 6.9 for rainfed, and in 2022 values ranged from from 5.0 to 7.5 for 100% ETc and from 5.0 to 6.3 for rainfed. In 2021, cumulative ETp ranged from 182.9 to 246.4 for 100% ETc and from 145.1 to 201.7 for rainfed. Cumulative ETa in 2021 varied from 242.9 to 321.1 for 100% ETc and from 199.1 to 272.3 for rainfed. In 2022 (the driest year), cumulative ETp ranged from 195.6 to 263.9 for 100% ETc and from 195.6 to 226.6 for rainfed. In 2022, cumulative ETa ranged from 220.9 to 306.5 for 100% ETc and from 175.8 to 209.6 for rainfed.

### 2.2. Exploring the Impact of Water Availability (in Irrigated and Rainfed Treatments) on Winter Wheat Varieties: Agronomic Performance, Cumulative LAI, and Evapotranspiration

The total amount of irrigation water applied throughout the growing season in the 100% ETc treatment was 234 and 270 mm for the first and second year, respectively (Figure 2). In addition, rainfall from sowing to physiological maturity was 112 mm and 148 mm for the first and second year, respectively (Figure 2). Soil water content measurements were performed in two wheat varieties (Variety 1 and Variety 19; Appendix A) which had shown contrasting behavior in previous studies (data not published). In the upper 40 cm of soil depth, the rainfed treatments exhibited notably lower soil water content in comparison with the 100% ETc treatment, with more pronounced differences observed at the maturity stage in both years. Remarkably, both wheat varieties experienced similar soil water conditions (see Appendix A), indicating similar water availability in the two varieties. An ANOVA was employed to assess the impact of year, irrigation treatment, variety, and their interactions with yield, water productivity (WP), LAI, total cumulative ETa, as well as cumulative ETa at both the vegetative and grain-filling stages (ETa VEG and ETa GF). Varietal influences were notable for yield, WP, LAI, and cumulative ETa VEG and GF. Moreover, total cumulative ETa remained consistent across all varieties. ETa VEG exhibited uniformity between the two irrigation treatments, while significant differences were observed for all other measured parameters (Table 2). Notably, interactions between irrigation treatment, variety, and year were statistically significant across most parameters, except for LAI and total cumulative ETa (Table 2). Varietal and irrigation treatment effects were assessed using mean separation tests (Table 3). The rainfed treatment resulted in a 30% reduction in grain yield (GY), a 52% reduction in cumulative ETa GF, and a 13% reduction in WP (Table 3). In terms of varietal effects, Variety 1 consistently demonstrated the highest grain yield (GY) under both irrigated and rainfed conditions, while Variety 19, Variety 20, Variety 21, and Variety 22 exhibited the lowest GY. Variability in ETa GF was notable (coefficient of variation 16.4%), with Variety 3 exhibiting the highest value and Variety 21 the lowest. Although ETa VEG was similar across the two irrigation treatments, there was substantial variation among varieties, with Variety 15 and Variety 22 recording the highest and lowest vales, respectively. In terms of WP, Variety 1 demonstrated the highest efficiency, while Variety 20 exhibited the lowest. Variety 1 emerged as a standout variety, characterized by high yields and relatively lower water consumption from cumulative ETa VEG and GF.

### 2.3. Correlation between Daily Cumulative Evapotranspiration Integral and Yield

The regression analysis encompassing all years, irrigation treatments, and wheat varieties revealed a significant and positive relationship between yield and total cumulative ETa, as depicted in Figure 2b. Subsequently, we delved deeper into the relative importance of cumulative ETa data from both the vegetative and grain-filling stages in influencing yield. Interestingly, cumulative ETa during the grain-filling stages emerged as the primary driver, supported by significant positive correlations observed between cumulative ETa GF and GY, as illustrated in Figure 3. Conversely, no such correlations were observed when considering cumulative ETa during the vegetative stage (Appendix A). The significance of evapotranspiration during the grain-filling stages was further underscored by the positive correlations observed between TKW and grain number with ETa GF (Figure 3). In contrast, such associations were not evident with ETa during the vegetative stage (ETa VEG, Appendix A). In all the functions we used, evapotranspiration approached an asymptote as yield, TKW, and grain number increased (Figure 2 and Figure 3). Lastly, it is noteworthy that these correlations, when considering the means of all varieties (genetic effects), were not universally significant, except for the positive correlation between WP and GY.

## 3. Discussion

### 3.1. Remote Sensing-Based Daily Cumulative LAI Integral

Commonly utilized models for daily LAI estimation encompass radiation-based paradigms exemplified by the simplified sun-canopy-sensor (S3) model, remote sensing-centric frameworks exemplified by MODIS, Sentinel-2 SPOT/VEGETATION LAI products [26], along with empirical constructs linking LAI with crop developmental phases. Machine learning methodologies, such as Random Forest regression and light interception models, also play pivotal roles. Model selection hinges upon the accessibility of data and the scale of investigation, with remote sensing models demonstrating efficacy at expansive scopes, while empirical models prove advantageous for more localized inquiries. UAV remote sensing data have gained prominence for deriving precise estimates of biophysical parameters including PH, f_c_, and VARI, as evidenced by Walter et al. [27] and Gitelson et al. [28,29], respectively. Parameters inferred from RGB imagery captured by UAVs can be judiciously amalgamated to compute LAI [30,31]. In our present investigation, we harnessed a multiple regression model, incorporating independent variables such as PH, f_c_ and VARI, to formulate a predictive equation (Equation (1)), facilitating LAI estimation for different wheat varieties across diverse image acquisition dates. This model yielded commendable results, evidenced by an RMSE of 0.46 and an R^2^ value of 0.85, denoting a robust alignment between estimated and observed LAI values. Notably, our observations indicate an ascending trajectory in VARI throughout the cropping season for both treatment groups, underscoring dynamic vegetation growth trends as expected.

### 3.2. Remote Sensing-Based Daily Cumulative Evapotranspiration Integral

Accurate assessments of actual evapotranspiration using the TSEB modelling scheme hold paramount significance within the realm of environmental science. These assessments play a pivotal role in addressing a spectrum of pressing concerns, encompassing the sustainability of agricultural ecosystems, the judicious management of water resources, and the refinement of selection criteria in plant breeding initiatives aimed at enhancing adaptability to arid and semi-arid environments. The escalating global demands for agricultural output, coupled with the escalating frequency of drought occurrences across diverse regions, underscore the imperative for a meticulous and comprehensive appraisal of the water demands of crops.

Presently, an array of tools and methodologies are at our disposal for the measurement or estimation of ETa at the field scale. These encompass instruments such as scintillometers [32], the eddy covariance (EC) technique [33,34,35], and the Bowen ratio–energy balance (BREB) measurement system [36,37]. Additionally, the Food and Agriculture Organization (FAO) crop coefficient methods [38] hold prominence, typically employed for estimating ETp at the field scale [39], relying on reference crop ET values derived from various models, as comprehensively reviewed by Djman et al. [40]. Moreover, certain models, including the Penman–Monteith model [41], the Shuttleworth–Wallace model [42], and the clumping model [43] are able to provide direct estimations of field-scale ETp. Within this realm, lysimeters [44,45] assume the role of a standard tool, routinely utilized for the assessment of field-scale approaches [46,47]. While these methods are readily accessible, challenges may arise during their implementation. For example, lysimeters, while valuable for studying evapotranspiration in controlled settings, can be challenging and time-consuming to use in the field due to factors such as installation, maintenance, and the need for environmental control. These complexities can limit their practicality for large-scale or long-term monitoring efforts. Therefore, it is crucial to develop simpler and more efficient tools and methodologies for the calculation of evapotranspiration. Streamlined measurement techniques and technologies can not only reduce the resource and time requirements but also enable broader and more accessible monitoring, facilitating a better understanding of water use in various ecosystems and aiding in sustainable water resource management. In line with this objective, remote sensing has been proposed as a tool to estimate ETa at any given time. The use of very-high-resolution thermal imagery, in combination with other approaches capable of estimating biophysical parameters of the vegetation such as PH, f_c_ and LAI, allows for the adoption of surface energy balance models and ETa estimation, as shown in the study by Bellvert et al. [25]. However, daily cumulative evapotranspiration is probably a more valuable metric than instantaneous measurements when assessing the water utilization of different wheat genotypes, especially in the context of crops with extended growth periods. This superiority stems from its ability to account for long-term water requirements, an essential consideration in the wheat cropping cycle. Wheat, like most crops, undergoes diverse growth stages (mainly vegetative and reproductive), each with unique water demands—from germination and vegetative growth to flowering, grain filling, and maturity. The vegetative stage of wheat is focused on developing leaves, stems, and roots to support future growth, with an emphasis on maximizing photosynthesis. In contrast, the reproductive stage shifts a plant’s energy toward producing flowers and seeds for propagation [48]. During this stage, wheat plants develop inflorescences, undergo pollination and fertilization, and concentrate on seed maturation and grain production. Relying on a single-day evapotranspiration measurement can fall short in capturing the evolving water needs throughout the crop’s lifecycle. In contrast, daily cumulative evapotranspiration provides a holistic perspective, encompassing water use over extended periods. Additionally, this cumulative approach facilitates equitable genotype comparisons. Wheat genotypes may exhibit varying water use efficiencies and growth patterns. Some genotypes may thrive during specific growth phases, while others maintain consistent water requirements throughout the growing season [49]. Hence, the methodologies developed here were specifically designed to estimate the daily cumulative evapotranspiration integral. In this study, we undertook an extensive examination of the seasonal evolution of LAI to derive potential and actual evapotranspiration integrals throughout the entire growth cycle of 22 distinct wheat varieties grown under both irrigated and rainfed conditions. Significantly, the evapotranspiration curves revealed notable variations, especially as the crop entered its reproductive phases. This observation corresponds to the prevailing conditions often witnessed in Mediterranean environments, characterized by diminishing precipitation and escalating temperatures, which pose a formidable challenge to crop development and the formation of grains.

### 3.3. Water Consumption, Crop Yield, and Water Productivity in Wheat Varieties Grown under Two Water Regime Systems (Full Irrigated and Rainfed)

Several studies have convincingly established a direct correlation between winter wheat ETa and crop yield [50]. In light of the constraints posed by limited water resources, as underscored by Bouras et al. [51], it becomes imperative to comprehend the variations in crop water consumption. In our current investigation, the discerned presence of an asymptotic relationship between ETa and yield-related traits (TKW and grain number) highlights a crucial point. There exists a threshold beyond which further augmentation of water availability or utilization may yield only marginal enhancement in crop yield or grain attributes. This observation suggests the existence of a yield plateau, particularly evident in cases of exceptionally high yields observed under full irrigation conditions. This insight holds significant implications for agriculture. It underscores the necessity of adopting resource-efficient farming practices and sustainable water management strategies. By recognizing the limits of the impact of water on crop productivity, agricultural stakeholders can make informed decisions aimed at optimizing resource use while maintaining ecological and economic sustainability.

It has been reported that grain filling is one of the most important crop stages contributing to yield under Mediterranean climates [52,53]. The use of the daily cumulative evapotranspiration integral in this study enabled estimation of potential water consumption for each wheat variety across different treatments, simplifying crop model adaptations to suit the unique characteristics of each variety. The results presented here show that while differences in the total daily cumulative evapotranspiration integral were not significant across the wheat varieties, significant differences were observed for the daily cumulative evapotranspiration during the grain filling and vegetative stages. Furthermore, the water supply remained consistent throughout the entire crop cycle for all wheat varieties. This was ensured by subjecting them to identical crop irrigation schedules and exposing them to the same amount of rainfall. However, the different varieties showed different phenological or developmental patterns with significant differences in the duration and timeframe of the vegetative and reproductive stages. This led to different patterns of water use through evapotranspiration in the different varieties. Moreover, the daily cumulative evapotranspiration integral at the vegetative stage was similar under irrigated and rainfed conditions, indicating that during the vegetative part of the crop cycle in the years tested no water limitations were evident at these stages. However, during grain filling, the daily cumulative evapotranspiration integral was reduced by around 52% (see Table 3), indicating clear water limitation during this stage of crop development. Although the total daily cumulative evapotranspiration integral did not significantly differ across wheat varieties, the distinct phenological patterns among these varieties led to variations in water use throughout the crop’s growth cycle. This highlights the complexity of crop–water interactions. To address these variations and optimize water-use efficiency and crop yield, there is a clear need for tailored irrigation strategies and varietal selection. This implies that agricultural practices should be adapted to suit specific wheat varieties and their developmental stages, particularly focusing on water management during grain filling to enhance overall crop productivity.

In the analysis of genetic variation across 22 wheat varieties, the examination of daily cumulative evapotranspiration during various growth stages revealed no significant correlations with crop yield. This suggests that genetic distinctions in how water is utilized during these growth stages appear to be independent of yield. This raises the prospect of a valuable opportunity: the potential to boost crop yield while concurrently reducing water consumption. Such an approach could enhance resource efficiency and sustainability, especially in semi-arid regions. Additionally, noteworthy differences were observed among wheat varieties regarding water productivity. Varieties exhibiting higher water productivity also demonstrated elevated grain yields. These findings imply the possibility of selecting wheat varieties can achieve increased grain production per unit area with the lowest amount of water use in the future that. Furthermore, the moderate yet promising reproducibility of water productivity, as measured by broad-sense heritability, suggests that selection of this trait has the potential to enhance yields in semi-arid regions. These results challenge the prevailing concept introduced by Dr. Blum [54], who posited that selection for high water-use efficiency under water-limited conditions would typically result in diminished yield and reduced drought resistance. These findings could be linked to the prevailing practice in previous studies, where water use has typically been calculated or estimated based on leaf traits, including transpiration efficiency and carbon isotope discrimination. The methods used in our study, particularly the calculation of the daily cumulative evapotranspiration integral, offer a more direct and encouraging approach to gauge water utilization and water productivity as an indicator of better performance under drought conditions.

## 4. Materials and Methods

### 4.1. Experimental Setup

The experiments were conducted in Catalonia (Spain) in the 2020–2021 (first year, 2021) and 2021–2022 (second year, 2022) growing seasons. During the first year the trial was located in Almacelles (41°43′54″ N, 0°25′24″ E, 221 m elevation), and in the second year in the location of Sucs (41°41′41″ N, 0°25′35″ E, 284 m elevation). The two sites are approximately 4 km apart and have a typical Mediterranean climate, with a mean annual rainfall of 461 mm and an annual reference evapotranspiration (ET_0_) of 1100 mm. First year rainfall and ET_0_ were 148 mm and 401 mm, respectively, while second year rainfall and ET_0_ were 112 mm and 402 mm, respectively, for both sites. Meteorological data were gathered from an automated weather station located approximately 3 and 5 km from the Almacelles and Sucs study sites, respectively. The weather station forms part of the official Catalonian network of meteorological stations (SMC, www.ruralcat.net/web/guest/agrometeo, accessed on 1 August 2023).

Twenty two winter wheat (*Triticum aestivum* L.) commercial varieties were evaluated under the following two contrasting treatments: (i) 100% ETc, irrigated at hundred percent of the potential crop evapotranspiration (ETc) throughout the growing season, and (ii) rainfed, which was not irrigated (Figure 4). Irrigation scheduling of both years was calculated with a water balance model, where the ETc was obtained as a product of the reference evapotranspiration (ETo) calculated with the Penman–Monteith [38] and crop coefficients (Kc) approach, which were derived from FAO-56 [38] and modified based on the weather conditions prior to being used for calculating ETc. Sprinklers were installed in the 100% ETc irrigation treatment in an 18 × 18 m grid and with a water flow discharge of 7.8 L/h/m^2^. Irrigation was scheduled once a week with a lateral Rainger sprinkler system.

In each treatment, varieties were planted following an incomplete block design with four replications and plots of 9.6 m^2^ (eight rows 8 m × 1.2 m and 0.15 m apart). Fields were maintained free of weeds, diseases, and pests using chemical treatments. The soil at the Almacelles study site had a clay texture and at Sucs a fine-loamy texture. Sowing was carried out on 14 December of 2020 and on 9 December of 2021, both at a density of 450 seeds/m^2^.

### 4.2. Image Acquisition Campaign

In both experimental years, thermal imagery was captured using a camera mounted on a DRONETOOLS HEXA-AG UAV (https://www.dronetools.es/, accessed on 1 August 2023) (Figure 5a). The thermal flights took place on 7 April (119 days after sowing, DAS), 19 April (131 DAS), and 28 May (170 DAS) in 2021, corresponding to the crop developmental stages of mid-jointing, around anthesis, and grain filling, respectively. In 2022, thermal flights were performed on 6 April (118 DAS), 26 April (138 DAS), and 13 May (155 DAS), corresponding to the crop developmental stages of jointing, around anthesis, and grain filling, respectively. The thermal camera used was a FLIR SC655 (FLIR Systems, Wilsonville, OR, USA) model with a spectral response in the range of 7.5–14.0 µm, a resolution of 640 × 480 pixels, a 6.8 mm focal length, and a field of view (FOV) of 45°.

Visible (RGB; red, green, and blue) images were captured with the UAV Typhoon H520E (Yuneec, https://www.us.yuneec.com/, accessed on 1 August 2023) (as shown in Figure 5b). RGB flights in the first year were conducted on 7 April (114 DAS), 19 April (126 DAS), and 28 May (165 DAS), while in the second year, flights occurred on 8 February (61 DAS), 28 February (81 DAS), 6 April (118 DAS), 26 April (138 DAS), and 13 May (155 DAS). The Typhoon H520E was equipped with a Yuneec E90X RGB camera (Yuneec, https://us.yuneec.com/e90/, accessed on 1 August 2023) featuring a resolution of 4864 × 3648 pixels, a 23 mm focal length, and a FOV of 91°. All flights were performed around 12:00 h (GMT) under clear sky conditions with wind speeds below 12 m·s^−1^. Flying height was settled to 50 m above ground level, resulting in a spatial resolution of 0.02 m for the RGB camera and 0.10 m for the thermal camera. Flight planning included 80% frontal and 60% side overlap. During the DRONEHEXA UAV image acquisition, in situ measurements were performed on various targets to correct the atmospheric influences on the signal. Temperature measurements were continuously taken on hot (black) and cold (white) targets, as well as in vegetation and bare soil, using the Calex thermal infrared radiometer (PC151LT-O, Pyrocouple series, Calex Electronics Limited, Bedfordshire, UK). Georeferencing was conducted using ground control points of which coordinates were acquired with a handheld Global Positioning System (Geo7x, Trimble GeoExplorer series, Sunnyvale, CA, USA). Orthomosaics were created using the Agisoft Metashape Professional software version 1.6.2 (Agisoft LLC., St. Petersburg, Russia). Radiometric and geometric corrections were performed with QGIS 3.28 (Quantum GIS 3.28.3) software.

### 4.3. Measurement of Soil Water Content

Soil water content was measured periodically using a neutron probe (Hydroprobe 503DR, CPN International, Inc., Martinez, CA, USA). A total of twelve 1.2 m access tubes were installed before the sowing date at the Variety 1 and Variety 19 variety micro-plots (three tubes per variety and treatment). These two varieties were chosen due to their contrasting yield response (Variety 1 provides consistently higher yields than Variety 19 in national post-registration trials). Measurements were taken in both study years at intervals of 20 cm depth on a total of 15 days in the period from March to June. Soil samples were taken for calibration using an auger with a removable sleeve and were weighed immediately. The samples were taken at 20 cm intervals down the soil profile and the dry mass content of each sample was measured after drying the soil to constant mass in a forced convection oven at 105 °C. The percentage of soil water content was then converted from a gravimetric to a volumetric measurement using a soil bulk density of 1.4 g·cm^−3^. Soil water depletion was calculated as the percentage difference between readings for two different dates at each depth.

### 4.4. Measurements of Agronomic Traits

Three replications per treatment were chosen to monitor crop development on a biweekly basis from the booting stage, with the intent of documenting the progress of growth stages (GS) [55]: GS55 (heading) and GS87 (physiological maturity). A specific developmental stage was considered to be achieved when roughly 50% of the plants exhibited the characteristic phenotypic features associated with that stage.

During each RGB UAV flight date, crop key biophysical variables were measured on the ground. The plant height (PH) measurements encompassed three main stems, from the tillering node to the top of the spike, excluding the awns. In one replication for each irrigation treatment, PH was measured in three individual plants within a single plot. Simultaneously, the leaf area index (LAI) was determined using a portable linear ceptometer (AccuPAR model LP-80, Decagon Devices Inc., Pullman, WA, USA). Measurements were conducted between 10:00 and 13:00 h (GMT) in one replicate of each irrigation treatment. Photosynthetically active radiation (PAR) beneath the wheat canopy was quantified by positioning the ceptometer horizontally at ground level and recording five PAR readings in each plot. The incident radiation above plants was retrieved using a fixed tripod connected to the sensor. Subsequently, LAI was estimated by using the LAI-calculator provided by AccuPAR-L80 (LAI-calculator, METER Group). Upon the culmination of the growth cycle, the plots were mechanically harvested at ripening, and crucial yield parameters were assessed, including grain yield at 13% moisture content (GY, kg·ha^−1^), thousand kernel weight (TKW, g), and grain number (grains·m^−2^).

### 4.5. Remote Sensing Estimates of Biophysical Variables

Remote sensing was used to estimate plant biophysical variables and crop ETa following the data acquisition and treatment flowchart shown in Figure 6. In this study, RGB images were used to estimate the biophysical traits (PH and f_c_). The VARI vegetation index (Equation (2)) was also used to assess crop seasonal development. RGB images are an option to consider when spectral information is not strictly required, as they can be acquired from compact UAVs that require a simpler field setup and image processing compared to custom setups of multispectral and thermal UAVs. The biophysical variables estimated from RGB images were combined with thermal data to estimate ETa using the TSEB model.

#### 4.5.1. VARI Vegetation Index

The VARI index was designed to highlight vegetation in the visible spectrum while mitigating illumination differences and atmospheric effects [29]. The VARI index was calculated from the resulting digital numbers (DN) of the RGB image as:(2)$$VARI=\frac{{DN}_{Green}-{DN}_{Red}}{{DN}_{Green}+{DN}_{Red}-{DN}_{Blue}},$$

#### 4.5.2. Plant Height

The three-dimensional PH was estimated by subtracting the digital terrain model from the digital surface model, which were both obtained from the photogrammetric point cloud of RGB images [21].

#### 4.5.3. Fractional Vegetation Cover

The fractional vegetation cover (f_c_) of each plot was also calculated by adapting the equation proposed by Gutman and Ignatov [56], and obtained as:(3)$$f_{c}=\frac{{VARI}_{i}-{VARI}_{soil}}{{VARI}_{veg}-{VARI}_{soil}},$$
where:VARI_i_ corresponds to the value on the vegetation;VARI_soil_ corresponds to the value of bare soil; andVARI_veg_ corresponds to the value of pure vegetation.

### 4.6. Seasonal Leaf Area Index (LAI)

A multiple regression model was developed to estimate the LAI using two biophysical variables (PH and f_c_) and the VARI vegetation index. First, calibration sets of plots were developed using 44 micro-plots (22 for 100% ETc and 22 for rainfed) on three dates when AccuPAR-L80 measures were retrieved (6 April, 26 April, and 13 May). The multiple regression model was constructed using 36 randomly selected plots of the 44. This was performed to ensure the plots used for calibration were different from those used in the model validation process. The best multiple regression model was selected based on the lowest Bayesian information criterion (BIC) in the calibration sub-set. The coefficient of determination (R^2^), equation parameters, and associated probabilities were calculated for the LAI model.

Once the LAI model had been developed, it was applied to the remaining plots for all dates. Subsequently, the seasonal evolution of LAI for each variety was developed as a function of DAS (Days After Sowing) and adjusted to a sigmoid logistic model as follows:(4)$${LAI}_{DAS}=\frac{c}{\left(1+Exp\left(-a\cdot \left(DAS-b\right)\right)\right)},$$
where:a = slope of the curve;b = growth steepest at midpoint; andc = asymptote.

### 4.7. Evapotranspiration and Water Productivity

ETp was estimated for each cultivar and date of the 2021 and 2022 growing seasons using the Penman–Monteith approach [38]. The daily modeled LAI for each cultivar from the previous section was used as input of the model.

For its part, ETa was estimated for each image acquisition date using the two-source energy balance (TSEB) model [18,57]. The TSEB modelling approach partitions the surface energy fluxes between nominal soil and canopy sources. In this study, both soil (T_S_) and canopy (T_C_) temperatures were retrieved from the high spatial resolution thermal images [17]. To obtain separately T_C_ and T_S_ for each variety, a two-step process was employed. Firstly, a supervised classification algorithm was applied on each image, effectively segregating the pixels into two categories: soil and vegetation. Subsequently, based on the obtained classification, the mean temperature values for vegetation and soil were computed for each individual plot. In addition to the surface temperatures, the model requires meteorological data and other biophysical inputs such as PH, LAI, and f_c_. Meteorological inputs were obtained from the weather station of the official Catalonian network of meteorological stations. For more information, the full python script is available online (https://github.com/hectornieto/pyTSEB, last accessed 13 August 2023).

Finally, crop stress coefficient (Ks) per image acquisition date was calculated as follows:(5)$$Ks=\frac{ETa}{ETp},$$

The Ks was used to interpolate ETa for the interflight periods (example given in Figure 7). At the first stages of the crop cycle (from 0 to 100 DAS), water needs were considered fulfilled without appreciable water restrictions, so that Ks was assumed to be equal to 1. The calculated Ks was then applied to each wheat variety as a function of flight date, obtaining the seasonal estimated ETa values. ETa was calculated for each variety for the whole crop cycle and also for the grain filling and vegetative periods by calculating cumulative evapotranspiration from sowing to heading (vegetative period) and from heading to maturity (grain filling period).

Once the daily ETa values had been estimated, the total sum was calculated for each growing season, for the vegetative period (ETa VEG, from sowing to the grain filling stage), and for the grain filling period (ETa GF). The lengths of the phenological stages of each variety were estimated visually through periodic field visits. Finally, the WP of each variety, defined as the amount of wheat grain produced per unit of evapotranspired water, was calculated as the ratio between crop yield and cumulative ETa (kg·m^−3^).

### 4.8. Statistical Analysis

Analyses of variances (ANOVAs) were conducted following a split-plot design. To prevent any potential cross-contamination between the irrigated and rainfed experimental conditions, specific randomization constraints were implemented. The primary factor under consideration was irrigation (fixed effect). Within each irrigation treatment, we applied randomization to the selection of wheat varieties (fixed effect) over two years (fixed effect), with three blocks per treatment (fixed effect). The error terms were designated as random effects and computed as follows: for irrigation effects, the Block*Irrigation interaction was utilized; for variety effects, it was Block*Variety*Irrigation; and for year effects, it was Block*Year*Irrigation*Variety. Means were compared using the Tukey test assuming a significance level of alpha = 0.05.

One way ANOVA and Fisher’s LSD post hoc tests were used in order to determine the effect of water regimes for all the analyzed variables. Linear regression equations and Pearson correlation coefficients were calculated to analyze the relationship between variables. Broad sense heritability (H^2^) was estimated for each trait individually in each environment and across all environments as:*H*^2^ = σ^2^g/[σ^2^g + (σ^2^/r)],(6)
where:r = number of repetitions;e = number of environments;σ^2^ = error variance;σ^2^g = genotypic variance; andσ^2^ge = genotype by environment interaction variance.

Analyses were conducted using the JMP 16.0 and SAS-STAT statistical package (SAS Institute Inc., Cary, NC, USA).

## 5. Conclusions

Accurate estimation of crop biophysical parameters was achieved through a combination of vegetation indices and photogrammetric analysis using RGB UAV data. The leaf area index (LAI) was estimated with high precision from regular RGB UAV flights, exhibiting a good accuracy (R^2^ > 0.85 and RMSE of 0.46).

Therefore, employing an interpolation-based methodology enabled the acquisition of valuable information of seasonal crop evapotranspiration and water status for each tested wheat variety. Notably, significant differences in cumulative evapotranspiration (ETa) were observed between treatments, as well as between varieties during the vegetative and grain-filling stages. The analysis encompassed the response of winter wheat evapotranspiration to drought stress throughout the entire crop cycle. Water productivity (WP) was also calculated.

The use of remote sensing for in-season daily LAI and Eta monitoring provides a comprehensive understanding of crop development in relation to environmental conditions. Daily estimates of vegetative development and ETa for different varieties have proven to be instrumental in selecting the most water-use efficient varieties. Interestingly, the varieties with the highest grain productivity did not necessarily exhibit the highest WP. Additionally, the performance of WP in irrigated varieties differed from those under rainfed conditions. Consequently, the plasticity of each variety in response to water stress, as measured by WP, is contingent upon the level of water scarcity. Hence, considering the degree of water restriction is crucial when selecting varieties based on WP functions.

## Figures and Tables

**Figure 1 plants-12-03871-f001:**
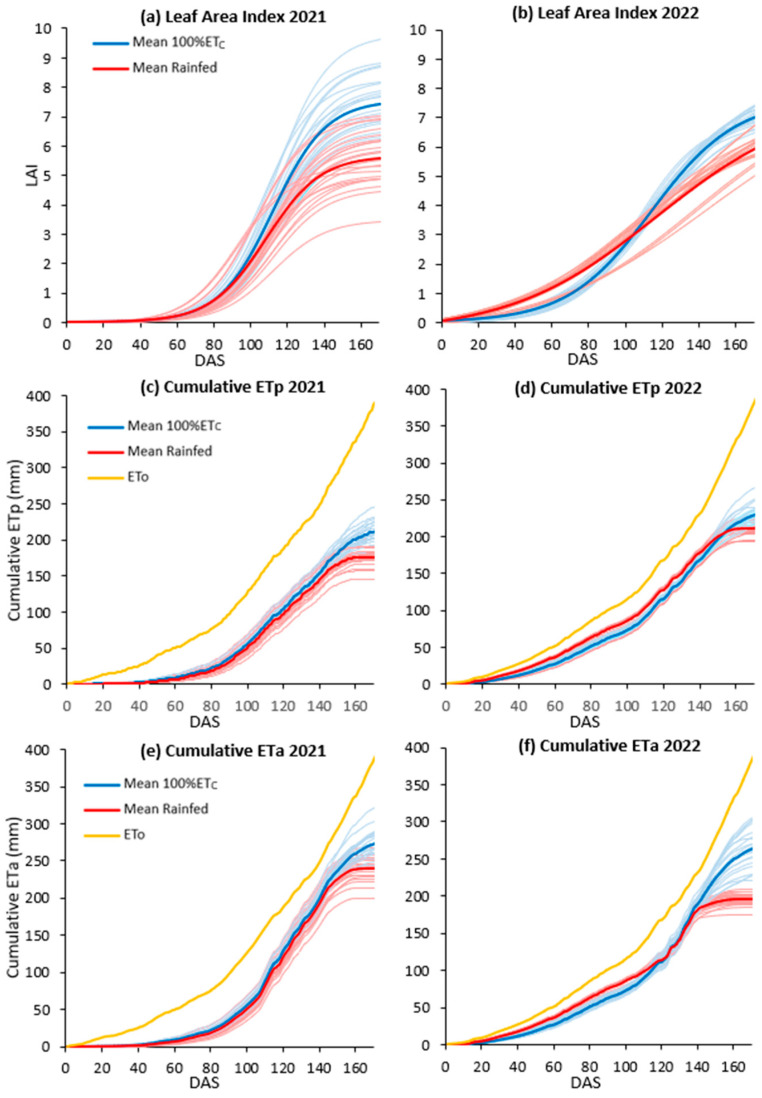
Seasonal evolution of leaf area index (LAI; (**a**,**b**)), cumulative potential evapotranspiration (ETp; (**c**,**d**)), and actual evapotranspiration (ETa; (**e**,**f**)). The data are presented in relation to Days After Sowing (DAS) for the 2021 and 2022 crop growing seasons. Blue corresponds to 100% ETc and red to rainfed treatments. Lighter lines correspond to the development of individual varieties and darker lines represent mean values per treatment.

**Figure 2 plants-12-03871-f002:**
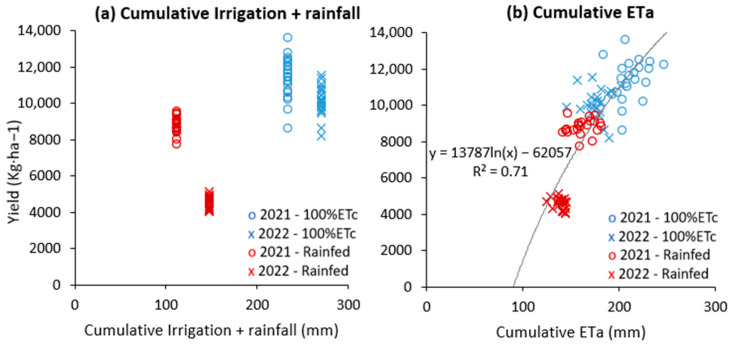
Relationship between yield and total amount of water applied (irrigation and rainfall (**a**)) and yield and cumulative actual evapotranspiration (**b**) for the 2021 and 2022 crop growing seasons.

**Figure 3 plants-12-03871-f003:**
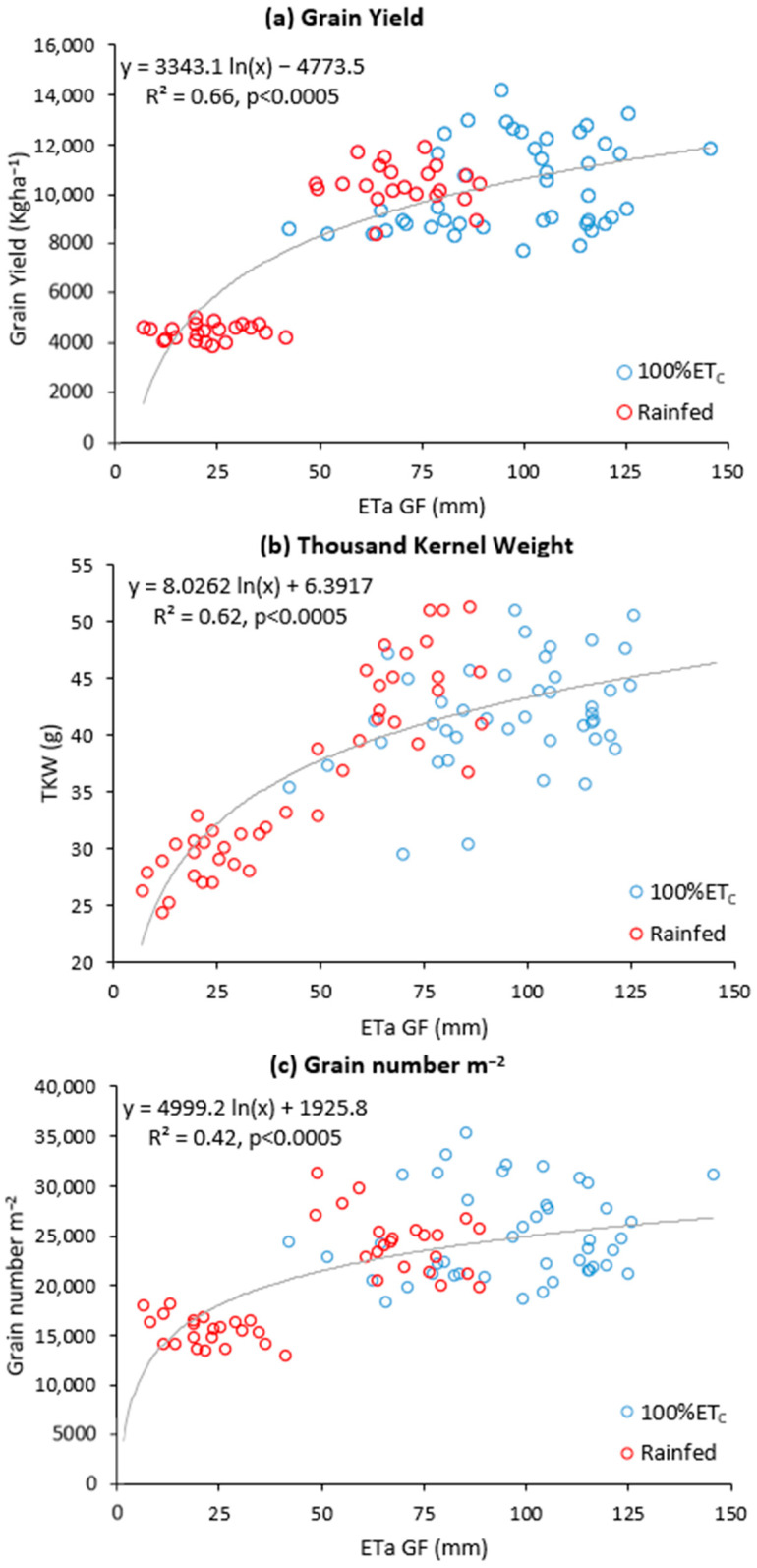
Relationships between evapotranspiration and (**a**) grain yield (kg ha^−1^), (**b**) thousand kernel weight (TKW, g), and (**c**) grain number per square meter during grain filling (data from two years, two water regimes, and 22 wheat varieties).

**Figure 4 plants-12-03871-f004:**
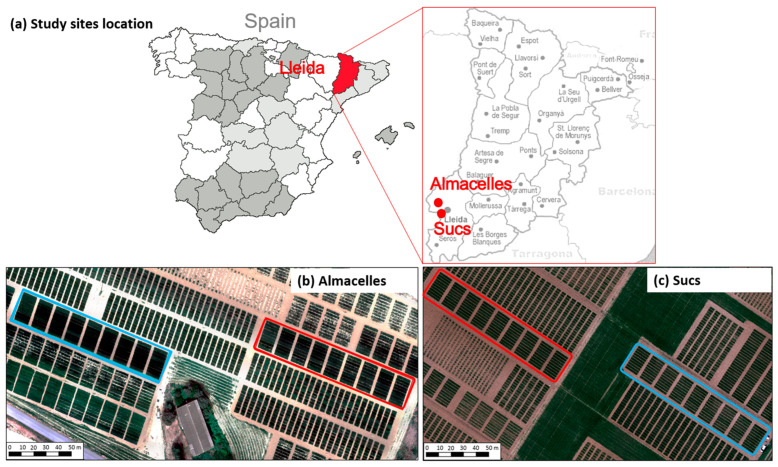
Location of study sites (**a**) Almacelles (**b**) and Sucs, and (**c**) experimental plots and layout of the field experiment with the two treatments: 100% ETc (blue) and rainfed (red).

**Figure 5 plants-12-03871-f005:**
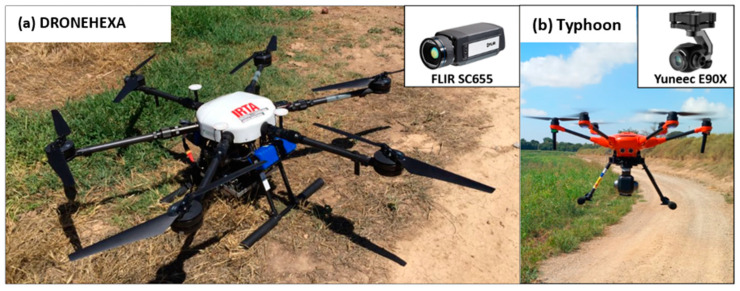
Unmanned aerial vehicles (UAVs) and cameras used in the study. DRONEHEXA UAV equipped with the thermal FLIR SC655 camera (**a**) and Yuneec Typhoon H520E equipped with a Yuneec E90X RGB camera (**b**).

**Figure 6 plants-12-03871-f006:**
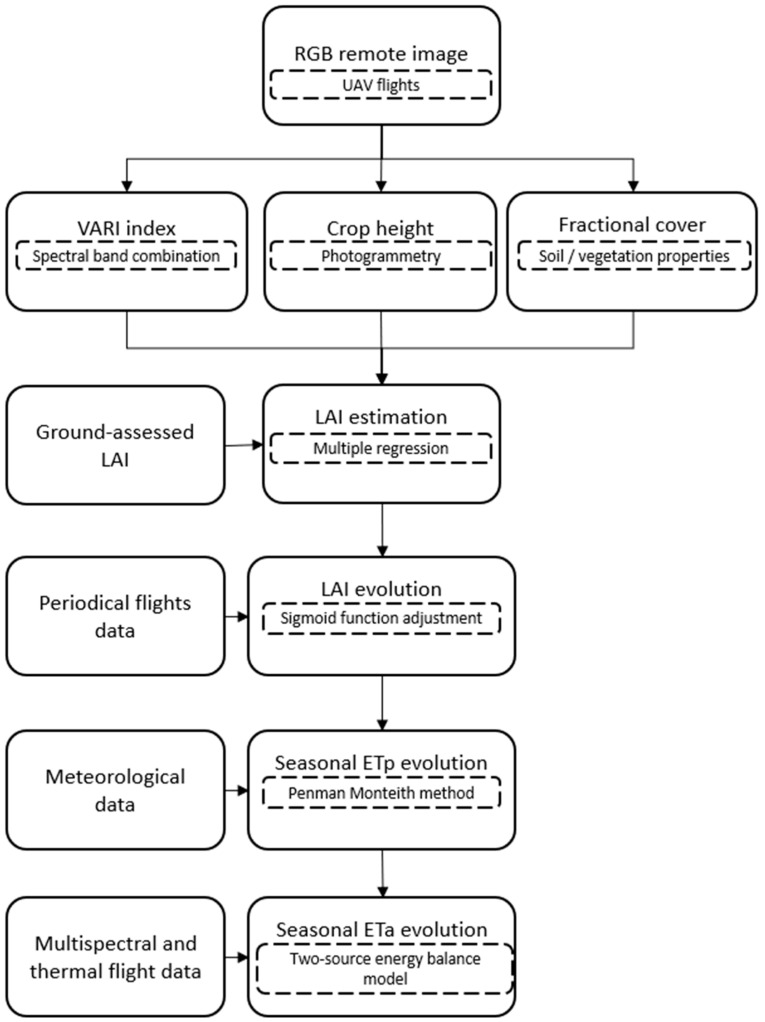
Flowchart of data acquisition and post-processing.

**Figure 7 plants-12-03871-f007:**
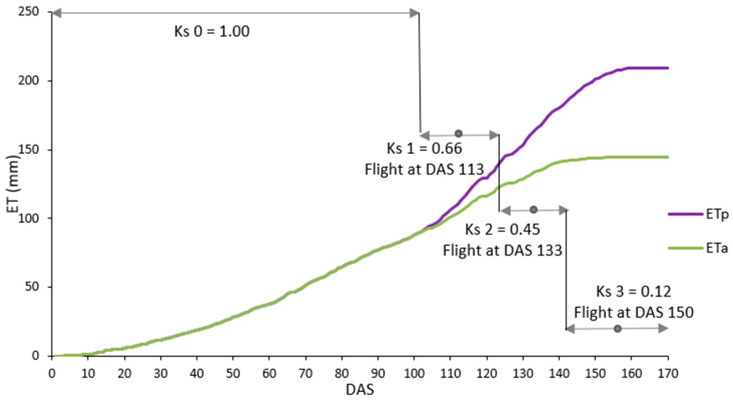
Example of estimates of the cumulative actual evapotranspiration (ETa) in a standard variety under rainfed conditions. Flight dates and intervals of the applied crop stress coefficients (Ks) are shown in grey. DAS = Days After Sowing.

**Table 1 plants-12-03871-t001:** Parameters of the multiple regression model to predict the leaf area index (LAI). The model was constructed with 180 plots (60 per flight date) to calculate multiple regression prediction equations by including the VARI (Visible Atmospherically Resistant Index) vegetation index, PH (plant height), and f_c_ (fraction of canopy cover) from remote sensing. Equation parameters are shown. The model was validated by calculating LAI using the equation obtained from multiple regression in 18 wheat trials with prediction accuracies (Pred Ac) shown as Pearson coefficients P(r) between observed LAI and predicted LAI.

Model	Estimation of Coefficients	Standard Error	Prob > |t|
INTERCEPT	−1.071	0.188	<0.0001 *
PH	5.063	0.274	<0.0001 *
VARI	5.451	0.766	<0.0001 *
f_c_	−1.661	0.440	0.0002 *
Summary of fit			
RMSE	0.505		
R^2^	0.829		
Pred Ac RF (Pearson r)	0.504		

* indicates statistically significant differences assuming a test significative level of 0.05.

**Table 2 plants-12-03871-t002:** Analysis of variance *p*-values (three-way ANOVA) testing the effects of year, treatment, and variety on leaf area index (LAI), cumulative actual evapotranspiration (ETa, mm), grain yield (GY, kg·ha^−1^), cumulative actual evapotranspiration during vegetative stages (ETa VEG, mm), cumulative actual evapotranspiration during grain filling (ETa GF, mm), and water productivity (WP, kg·m^−3^). Na: Not available. For the cumulative actual evapotranspiration, data from all three blocks were used for calculation and all block effects were removed from the ANOVA.

Source		LAI	ETa	GY	ETa VEG	ETa GF	WP
	Nparm	Prob > F	Prob > F	Prob > F	Prob > F	Prob > F	Prob > F
Block	2	0.8693	Na	0.2495	0.9077	0.9077	0.2053
Irrigation	1	0.8367	<0.0001 *	0.0049 *	0.6107	0.0004 *	0.0198 *
Variety	21	0.0500 *	<0.0001 *	<0.0001 *	<0.0001 *	<0.0001 *	<0.0001 *
Irrigation*Variety	21	0.4215	0.999	0.0015 *	<0.0001 *	<0.0001 *	<0.0001 *
Variety*Block	42	0.4908	Na	0.1283	0.6513	0.6513	0.0928
Year	1	0.0005 *	<0.0001 *	<0.0001 *	<0.0001 *	<0.0001 *	<0.0001 *
Year*Irrigation	1	0.028 *	<0.0001 *	<0.0001 *	0.0264 *	<0.0001 *	<0.0001 *
Year*Variety	21	0.1889	0.0007 *	0.0116 *	<0.0001 *	<0.0001 *	0.0148 *
Year*Irrigation*Variety	21	0.3597	<0.0001 *	0.0375 *	<0.0001 *	<0.0001 *	0.0139 *
Year*Block	2	0.6443	Na	0.4697	0.0052 *	0.0052 *	0.5735 *
Year*Irrigation*Block	2	<0.0001 *	Na	0.0008 *	0.0086 *	0.0086 *	0.0008 *
Year*Variety*Block	42	0.7803	Na	0.7536	0.8793	0.8793	0.7799

* indicates statistically significant differences assuming a test significative level of 0.05.

**Table 3 plants-12-03871-t003:** Leaf area index (LAI), cumulative actual evapotranspiration (ETa), mean grain yield (GY, kg·ha^−1^), cumulative evapotranspiration during grain filling (ETa GF), and the vegetative stages (ETa VEG) and water productivity (WP) for the twenty-two wheat varieties grown for two years and under two water regimes. Broad-sense heritability (H^2^) is shown for the mean comparison. Pearson’s correlation coefficient (Correlation P GY) of ETa GF, ETa VEG, and WP with grain yield) is also shown.

Variety	LAI	ETa	GY	ETa GF	ETa VEG	WP
		mm	kg ha^−1^	mm	mm	kg Grain m^−3^
Variety 1	6.32	232.0	9951.1 a	68.9 fgh	163.1 ij	4.2 a
Variety 2	6.63	241.4	9356.9 ab	67.0 ghi	174.4 efg	3.8 abcd
Variety 3	5.93	248.1	9330.4 ab	86.7 a	161.4 jk	3.7 abcde
Variety 4	6.93	270.5	9238.3 abc	83.3 abcde	187.2 b	3.4 bcde
Variety 5	6.32	234.4	9215.6 abc	66.5 ghij	167.9 hi	3.9 ab
Variety 6	6.76	257.9	9154.2 abcd	78.2 e	179.7 cde	3.5 bcde
Variety 7	6.10	234.8	9109.3 abcd	71.4 fg	163.4 ij	3.8 abc
Variety 8	6.50	246.1	9108.7 abcd	61.5 j	184.6 bc	3.6 bcde
Variety 9	6.84	255.7	9105.8 abcd	84.6 ab	171.1 fgh	3.5 bcde
Variety 10	5.66	230.3	9011.5 abcd	80.5 bcde	149.8 m	3.8 ab
Variety 11	6.50	236.8	9002.6 abcd	64.0 hij	172.8 fgh	3.7 abcd
Variety 12	6.91	252.0	8919.5 abcd	72.7 f	179.3 cde	3.4 bcde
Variety 13	6.16	242.4	8875.9 abcd	71.7 fg	170.7 gh	3.6 bcde
Variety 14	7.22	252.9	8862.2 abcd	79.1 cde	173.8 fg	3.4 bcde
Variety 15	7.15	250.2	8783.7 abcd	54.9 k	195.3 a	3.4 bcde
Variety 16	6.63	263.8	8762.3 abcd	52.9 k	180.9 cd	3.7 abcde
Variety 17	5.60	236.7	8730.9 abcd	82.5 abcde	154.2 lm	3.6 abcde
Variety 18	6.51	220.4	8718.9 abcd	62.7 ij	157.7 kl	3.9 ab
Variety 19	6.38	240.8	8336.9 bcd	84.4 abc	156.4 kl	3.4 bcde
Variety 20	6.99	255.4	8122.2 bcd	79.0 de	176.4 def	3.2 e
Variety 21	7.23	256.2	7983.2 cd	43.4 l	197.8 a	3.2 de
Variety 22	6.06	236.9	7869.9 d	83.7 abcd	153.2 lm	3.3 cde
** *100% ETc* **	7.22 A	268.3 A	10,323.9 A	97.2 A	171.1 A	3.8 A
** *Rainfed* **	5.81 A	218.1 A	7453.3 B	46.4 B	171.7 A	3.3 B
**H^2^**			0.4900	0.6444	0.8890	0.5593
**Correlation P GY**				0.091	−0.045	**0.742**

Different letters, whether in uppercase or lowercase, indicate significant differences assuming a significance level of alpha = 0.05, using Tukey’s fair significant difference test.

## Data Availability

Data are contained within the article and Appendix A.

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
