# Peer review of "A Remote Sensing Approach for Assessing Daily Cumulative Evapotranspiration Integral in Wheat Genotype Screening for Drought Adaptation"

_plants, 2023, doi:10.3390/plants12223871_

Round 1

Reviewer 1 Report

Comments and Suggestions for Authors

Due to the growing global requirements for agricultural production, combined with the increasing frequency of droughts in various regions of the world, the authors of the work undertook a very important task. Emphasizing the need for a meticulous and comprehensive assessment of crop water demand, they conducted a very extensive study and obtained an accurate estimate of crop biophysical parameters by combining vegetation indices and photogrammetric analysis using RGB UAV data. Although the work constitutes an important contribution to the literature on the subject, some flaws were not avoided:

lines 18-19. Please explain what you mean by „positive asymptotic trend”?

line 118: VARI – what does this abbreviation mean??

Line 123: what does the abbreviation mean: fc and below RMSE (line 126). The description of markings must appear at the first appearance of a given marking in the text of the work.

In line 127 you wrote: „Equation 5 was employed to calculate the LAI for each variety and image acquisition date.” Meanwhile, formula (5) describes: crop stress coefficient. Can you verify this?

Line 131: RMSE 0.02 m – the figure shows RMSE = 0.01 m – which is correct?

Line 133. ETp designation not explained

Line 154: I suggest writing it in a more correct form, i.e.: * means statistically significant differences assuming a test significance level of 0.05

I consider line 155 unnecessary, especially since you do not precisely specify information about: x1, x2, x3.

In table 1, replace "estimation" with "estimation of coefficients"

Line 158: no info about s.e.

Please think about the numbering of the drawings. The current one is questionable... (I see Figure 5 but I don't see Figures 1-4)

The description of Table 2 does not indicate what values Table 2 presents. Furthermore, Table 2 is incomplete. You refer to the symbol "*", which is missing in table 2. Additionally, replace line 174 similarly to line 154. The current notation is not correct.

In my opinion, the order in which the information is placed is not correct. For example, you put Table 2 first and only refer to it in the next paragraph. I think you should change this order!

Line 184. You wrote: Soil water content measurements were performed in two contrasting wheat varieties (Marcopolo and Nogal; Supplementary Figure 2) – please elaborate on why you considered these varieties to be contrasting

Line 203 - you write "Variability in ETa GF was notable..." - I think that for justification you should provide the value of the coefficient of variation.

Why did you use the same markings for the varieties and for 100%ETc and Rainfed. Shouldn't other symbols appear in the last two cases, e.g. capital letters of the alphabet?

Replace line 216 with: Different letters indicate significant differences assuming a significance level of alpha = 0.05, using Tukey's fair significant difference test

Include in the supplementary material a correlation table justifying the description of paragraph 2.3.

Line 583: Replace the sentence: Means were compared with a Tukey test at P<0.05” with a more reasonable one: Means were compared using the Tukey test assuming a significance level of alpha = 0.05.

In lines 583 – 587 there was an unexpected repetition of information.

Author Response

Dear reviewer 1,

I am pleased to have an opportunity to make the revision of my manuscript entitled “A Remote Sensing Approach for Assessing Daily Cumulative Evapotranspiration Integral in Wheat Genotype Screening for Drought Adaptation”.

In revised manuscript, I have carefully considered your comments. I reply to each comment in point-by-point fashion.

You will find below this letter detailed responses to comments/remarks.

In the updated version of the manuscript, we have substituted all mentions of the proprietary names of the varieties with generic designations (genotypes 1 to 22). This modification was made in accordance with the restrictions imposed by commercial companies, which prohibit the disclosure of their registered genotype commercial names in scientific publications.

Yours Sincerely

   David Gómez-Candón

------------------------------------------------------------------------------

Reviewer 1:

Due to the growing global requirements for agricultural production, combined with the increasing frequency of droughts in various regions of the world, the authors of the work undertook a very important task. Emphasizing the need for a meticulous and comprehensive assessment of crop water demand, they conducted a very extensive study and obtained an accurate estimate of crop biophysical parameters by combining vegetation indices and photogrammetric analysis using RGB UAV data. Although the work constitutes an important contribution to the literature on the subject, some flaws were not avoided:

lines 18-19. Please explain what you mean by „positive asymptotic trend”?

Author comment:

The meaning of “positive asymptotic trend” was: The amount of water evapotranspired by crops has a direct relationship with yield. However, this constant increase approaches a positive maximum value without ever reaching it, indicating a steady but limited increase in crop yield in relation with ETa.

I rewrote the sentence to clarify it: “An examination of the relationships between crop yield and evapotranspiration (ETa), while considering factors like year, irrigation methods, and wheat cultivars, unveiled a pronounced positive asymptotic pattern”.

line 118: VARI – what does this abbreviation mean??

Author comment: “VARI (Visible Atmospherically Resistant Index)” added to the text.

Line 123: what does the abbreviation mean: fc and below RMSE (line 126). The description of markings must appear at the first appearance of a given marking in the text of the work.

Author comment: Changed, thank you for your comment.

In line 127 you wrote: „Equation 5 was employed to calculate the LAI for each variety and image acquisition date.” Meanwhile, formula (5) describes: crop stress coefficient. Can you verify this?

Author comment: It was Equation 1. In the new version of the manuscript is corrected.

Line 131: RMSE 0.02 m – the figure shows RMSE = 0.01 m – which is correct?

Author comment: The correct value was 0.01. Changed.

Line 133. ETp designation not explained

Author comment: “potential evapotranspiration (ETp)” added to the text.

Line 154: I suggest writing it in a more correct form, i.e.: * means statistically significant differences assuming a test significance level of 0.05

Author comment: Changed in both Table 1 and Table 2.

I consider line 155 unnecessary, especially since you do not precisely specify information about: x1, x2, x3.

Author comment: I agree, deleted.

In table 1, replace "estimation" with "estimation of coefficients"

Author comment: Ok, replaced

Line 158: no info about s.e.

Author comment: “standard error” added to the text.

Please think about the numbering of the drawings. The current one is questionable... (I see Figure 5 but I don't see Figures 1-4)

Author comment: In the new version of the manuscript figures are re-numbered.

The description of Table 2 does not indicate what values Table 2 presents. Furthermore, Table 2 is incomplete. You refer to the symbol "*", which is missing in table 2. Additionally, replace line 174 similarly to line 154. The current notation is not correct.

Author comment: Table 2 and its description updated.

In my opinion, the order in which the information is placed is not correct. For example, you put Table 2 first and only refer to it in the next paragraph. I think you should change this order!

Author comment: I agree. Table 2 moved.

Line 184. You wrote: Soil water content measurements were performed in two contrasting wheat varieties (Marcopolo and Nogal; Supplementary Figure 2) – please elaborate on why you considered these varieties to be contrasting

Author comment: Sentence updated: “Soil water content measurements were performed in two wheat varieties (Marcopolo and Nogal; Supplementary Figure 2) which had shown contrasting behavior in previous studies (data not published)”.

Line 203 - you write "Variability in ETa GF was notable..." - I think that for justification you should provide the value of the coefficient of variation.

Author comment: “coefficient of variation 16.4%” added to the text.

Why did you use the same markings for the varieties and for 100%ETc and Rainfed. Shouldn't other symbols appear in the last two cases, e.g. capital letters of the alphabet?

Author comment: Thank you for your comment. I changed to capital letters as you suggested.

Replace line 216 with: Different letters indicate significant differences assuming a significance level of alpha = 0.05, using Tukey's fair significant difference test

Author comment: Ok, done.

Include in the supplementary material a correlation table justifying the description of paragraph 2.3.

Author comment: Correlation table added as “Supplementary Table 1”

Line 583: Replace the sentence: Means were compared with a Tukey test at P<0.05” with a more reasonable one: Means were compared using the Tukey test assuming a significance level of alpha = 0.05.

Author comment: Thank you for your comment. Done.

In lines 583 – 587 there was an unexpected repetition of information.

Author comment: Corrected.

Reviewer 2 Report

Comments and Suggestions for Authors

Comments to Authors:

1.      Line 34: the word "sector" is repeated twice in the sentence. Please delete one.

2.      The references in the text were written using full names and numbers. Please ensure that you adhere to the citation guidelines of the journal when citing references in the text.

3.       Numerous studies have been conducted to evaluate the spectral assessment of drought tolerance in wheat genotypes under arid conditions. However, the authors of this study did not incorporate any of these previous studies. Here is an example:

a.        Hyperspectral reflectance sensing to assess the growth and photosynthetic properties of wheat cultivars exposed to different irrigation rates in an irrigated arid region. PLoS One 12 (8), e0183262.

4.      The placement of Supplementary Figure 1 in the manuscript is incorrect. It should be placed in a separate file.

5.      Line 127: Equation No. 5 is not related to calculating the LAI for each variety and image acquisition date, but rather to calculating Ks (please check M&M section).

6.      Line 132: Because the Results section has been placed before the Materials and Methods section, the tables in the text must be renumbered accordingly.

7.      Lines 192-193: Please provide the full name for any abbreviations that have been mentioned for the first time in the manuscript.

8.      Line 226: Figure 6b represents a regression analysis, not a correlation analysis.

9.      Lines 401-402: Please include the weather data for both locations in the M&M section.  

10.  Lines 404-407: Could you please provide the pedigree of the genotypes used in the study?

11.  Please confirm whether the genotypes are winter wheat or spring wheat.

12.  Lines 412-413: Is the Kc of FAO-56 modified based on the weather conditions of the experiment locations prior to being used for calculating ETC?

13.  Lines 414-415: If irrigation is scheduled once a week, what is the impact of rainfall on the amount of water needed for the 100%ETc treatment during the growing season?

14.  Line 582: Statistical analysis necessitates the expertise of a specialist due to the experiment being conducted in two distinct locations and the data being analyzed using a split-plot design. Therefore, it is imperative to seek consultation from a statistical analysis specialist prior to granting approval for the publication of this research.

Author Response

Dear reviewer 2,

I am pleased to have an opportunity to make the revision of my manuscript entitled “A Remote Sensing Approach for Assessing Daily Cumulative Evapotranspiration Integral in Wheat Genotype Screening for Drought Adaptation”.

In revised manuscript, I have carefully considered your comments. I reply to each comment in point-by-point fashion.

You will find below this letter detailed responses to comments/remarks.

In the updated version of the manuscript, we have substituted all mentions of the proprietary names of the varieties with generic designations (genotypes 1 to 22). This modification was made in accordance with the restrictions imposed by commercial companies, which prohibit the disclosure of their registered genotype commercial names in scientific publications.

Yours Sincerely

   David Gómez-Candón

------------------------------------------------------------------------------

 Reviewer2:

  1. Line 34: the word "sector" is repeated twice in the sentence. Please delete one.

Author comment: “sector” word repetition corrected.

  1. The references in the text were written using full names and numbers. Please ensure that you adhere to the citation guidelines of the journal when citing references in the text.

Author comment: As you suggested, in the new version of the manuscript references were presented following journal requirements.

  1. Numerous studies have been conducted to evaluate the spectral assessment of drought tolerance in wheat genotypes under arid conditions. However, the authors of this study did not incorporate any of these previous studies. Here is an example:
  2.  Hyperspectral reflectance sensing to assess the growth and photosynthetic properties of wheat cultivars exposed to different irrigation rates in an irrigated arid region. PLoS One 12 (8), e0183262.

Author comment: As you suggested, 2 new references were added to the text.

  1. The placement of Supplementary Figure 1 in the manuscript is incorrect. It should be placed in a separate file.

Author comment: In the new version both 1 and 2 Supplementary Figures were placed in a separate file.

  1. Line 127: Equation No. 5 is not related to calculating the LAI for each variety and image acquisition date, but rather to calculating Ks (please check M&M section).

Author comment: It was Equation 1. In the new version of the manuscript is corrected.

  1. Line 132: Because the Results section has been placed before the Materials and Methods section, the tables in the text must be renumbered accordingly.

Author comment: In the new version tables and figures numbers are corrected.

  1. Lines 192-193: Please provide the full name for any abbreviations that have been mentioned for the first time in the manuscript.

Author comment: Changed, now every of the abbreviations in the text are mentioned correctly.

  1. Line 226: Figure 6b represents a regression analysis, not a correlation analysis.

Author comment: Changed.

  1. Lines 401-402: Please include the weather data for both locations in the M&M section.  

Author comment: Weather data added. It is remarkable that both locations were 4000 m apart, so weather data was equal for the two.

  1. Lines 404-407: Could you please provide the pedigree of the genotypes used in the study?

Author comment: Due to restrictions imposed by commercial companies, we are unable to disclose the registered commercial names or pedigrees of the genotypes used in this study. Consequently, in the revised version of the manuscript, all references to the commercial names of these varieties have been substituted with generic designations (genotypes 1 to 22).

  1. Please confirm whether the genotypes are winter wheat or spring wheat.

Author comment: Winter wheat genotypes were used in the experiment. This point is now it is clarified in the new version of the text.

  1. Lines 412-413: Is the Kc of FAO-56 modified based on the weather conditions of the experiment locations prior to being used for calculating ETC?

Author comment: They were modified prior to being used for calculating ETc. It is specified in the new version of the text.

  1. Lines 414-415: If irrigation is scheduled once a week, what is the impact of rainfall on the amount of water needed for the 100%ETc treatment during the growing season?

Author comment: The previous week rainfall amount was considered to be added to the soil water reservoir. Therefore, it was subtracted from the 100%ETc estimated water input for the week.

  1. Line 582: Statistical analysis necessitates the expertise of a specialist due to the experiment being conducted in two distinct locations and the data being analyzed using a split-plot design. Therefore, it is imperative to seek consultation from a statistical analysis specialist prior to granting approval for the publication of this research.

Author comment: The statistical analyses have been reviewed by an expert. The tables have been updated according to the expert's guidance. The description of the statistical analyses carried out has also been updated in the material and methods section.

It should be noted that an error was detected in the ANOVA calculation for the ETa. In the new version of the article this error has been corrected.

Round 2

Reviewer 1 Report

Comments and Suggestions for Authors

I don't have any other suggestions.

Regards

Reviewer 2 Report

Comments and Suggestions for Authors

Please keep the names of the varieties. This is important for other research studies.